# Strengthening the Neonatal Workforce Through World Café Methodology

**DOI:** 10.3390/healthcare13222976

**Published:** 2025-11-19

**Authors:** Suza Trajkovski, Linda Ng, Patricia Lowe, Margaret Broom

**Affiliations:** 1School of Nursing and Midwifery, Penrith Campus, Western Sydney University, Penrith, NSW 2751, Australia; s.trajkovski@westernsydney.edu.au; 2School of Nursing & Midwifery, University of Southern Queensland, 11 Salisbury Road, Ipswich, QLD 4305, Australia; 3Australian College of Nurses, Level 6, 9 Wentworth Street, Parramatta, NSW 2150, Australia; patricia.lowe@acn.edu.au; 4Canberra Hospital, Canberra Health Services Yamba Dr, Garran, ACT 2605, Australia; margaret.broom@act.gov.au

**Keywords:** neonatal, neonatal intensive care, nursing, special care nursery, World Café methodology, workforce

## Abstract

**Highlights:**

**Impact:** This study contributes valuable insights into addressing potential neonatal nursing shortages. It provides guidance for recruitment efforts, informs policy decisions, and helps prepare for the future neonatal workforce. **Reporting Method:** Compliance with the SRQR guidelines for reporting qualitative research was maintained. **Patient or Public Contribution:** No patient or public contribution. **What Does this Article Contribute to the Wider Global Clinical Community?** This study found that Nursing Associations, Nurse Managers, and Clinical Nurses have a shared responsibility to strengthen advocacy, leadership, professional identity, innovation, and support within neonatal nursing. **Recommendations to enhance both workforce satisfaction and patient care:** This study highlights the growing need to identify the specific training requirements of neonatal nurses to better understand their transition needs. Notably, mentorship plays a crucial role in their professional development and advancement at all levels. It also serves as a vital countermeasure against burnout and moral distress.

**Abstract:**

**Aim:** This workshop aimed to facilitate in-depth discussions, promote knowledge sharing, and generate evidence-informed strategies to improve the recruitment and retention of the neonatal nursing workforce in Australia. **Research Design:** A qualitative participatory study was conducted using the World Café methodology to engage neonatal nurses in collaborative dialogue. **Methods:** Twenty-seven neonatal nurses (all female, predominantly aged 40–60 years) participated in a structured World Café workshop. An inductive thematic analysis was employed to explore participants’ perspectives on workforce challenges and solutions. **Results:** Four key themes were identified: (1) Leadership and Advocacy, (2) Professional Identity, (3) Future Vision and Innovation, and (4) Creating a Supportive Culture. These themes reflect the complex, interrelated factors influencing workforce sustainability and highlight the need for targeted, multi-level interventions. **Conclusions:** Building a resilient neonatal nursing workforce and ensuring the delivery of high-quality care requires a coordinated response from professional nursing organisations, nurse leaders and managers, and individual clinicians. This study provides a replicable model for stakeholder engagement and offers actionable recommendations to inform workforce policy, leadership development, and practice innovation.

## 1. Introduction

Globally, the female-dominated nursing and midwifery workforce is in crisis. Labour market forces, such as secondary and tertiary graduation figures and unmatched net migration figures, lead to workforce imbalances [1]. The World Health Organization [2] estimates a shortage of 4.5 million nurses and 0.31 million midwives by 2030, with the largest shortfalls anticipated in countries such as Africa, Southeast Asia, Latin America, and the Eastern Mediterranean region. Workforce deficits are also evident in Australia.

Australian nurses and midwives comprise approximately 40% of the national health workforce [3]. Although workforce data from the Australian Department of Health and Aged Care (2018–2022) show an overall increase in registered and employed nurses, enrolled nurses, and midwives, there has been a decline in dual registrations and a notable inverse relationship between workforce age and average hours worked [4]. For instance, older workforces in South Australia report fewer average hours worked compared to younger cohorts in the Australian Capital Territory and Northern Territory [4]. These trends signal emerging workforce challenges, including anticipated shortages and skill mix imbalances as the workforce ages [3]. This is particularly concerning in neonatal nursing, where shortages have been documented and may impact care delivery and workforce sustainability [5].

Despite graduating more nurses per 100,000 population (115.7) than any other Organisation for Economic Co-operation and Development (OECD) country [6], Australia faces persistent challenges in aligning nursing and midwifery skills with the evolving health needs of its diverse population. These challenges are compounded by inconsistent funding structures and the absence of a coordinated national nursing workforce policy [3], which limit strategic workforce planning, particularly in specialised areas such as neonatal nursing. Australia’s nursing and midwifery workforce reflects broader demographic trends, with a predominance of urban-dwelling and culturally and linguistically diverse practitioners [7,8]. Nearly three-quarters of the population resides in urban areas [7], and the proportion of individuals aged 85 years and older is projected to rise significantly [8]. These demographic shifts, combined with uneven resource allocation, place increasing pressure on neonatal services, which require highly skilled staff to meet complex care demands across both metropolitan and regional settings.

Approximately 25% of the Australian population speak a language other than English at home [8], reflecting the cultural and linguistic diversity of the broader health workforce. While this diversity enriches care delivery, it also presents challenges in neonatal units, where effective communication and culturally sensitive care are critical to supporting families during vulnerable periods. Preparing nurses and midwives to meet these needs is further complicated by fragmented state and territory-specific funding models and inconsistent recognition of skills across jurisdictions [9]. Additionally, the post-pandemic workforce faces significant strain, with rising levels of burnout, impaired wellbeing, and increased exposure to occupational violence [8]. These factors collectively impact neonatal nursing capacity, where continuity of care, emotional resilience, and specialised communication skills are essential to delivering safe and responsive care.

Currently, both anecdotal and empirical evidence suggest that the wellbeing of Australian nurses and midwives is significantly compromised. Wellbeing, defined as the balance between contentment and distress, directly influences overall health, professional engagement, and quality of life [8]. During the COVID-19 pandemic, organisational disruptions and staffing shortages intensified role strain, contributing to burnout, fear, and diminished morale [10,11]. These experiences have led to increased intentions to leave the profession, raising concerns about long-term workforce sustainability [9]. In response, healthcare leaders in Australia and New Zealand surveyed the workforce and developed a set of strategic recommendations to support nursing staff [12], highlighting the urgent need for coordinated, evidence-based national approaches to education, wellbeing, and retention. One promising initiative is the Australian Health Department’s strategy to improve workforce sustainability, diversity, and planning through innovations such as data sharing and standardised career pathways [3]. These efforts are particularly critical for neonatal services, where workforce stability and emotional resilience are essential to maintaining high-quality, family-centred care.

Accessing a sufficient number of skilled nurses and midwives to deliver specialised, technologically advanced care remains a persistent challenge across Australia. One critical area is neonatal care, which refers to the care of newborns within the first 28 days of life. Neonatal nurses, registered nurses and midwives with specialised training, provide care to infants born prematurely or with medical or surgical conditions in neonatal care units [13]. Demand for neonatal services remains steady, with over 8% of live-born Australian babies born prematurely and approximately 17% requiring admission to a newborn care facility [14]. Despite this consistent demand, workforce data specific to neonatal nursing is limited. The most recent figures, collected between 2016 and 2019, showed a modest increase in nurses and midwives identifying neonatal care as their clinical specialty [15]. However, these figures also revealed that fewer than 2% (*n* = 5349) of the national healthcare workforce possessed the specialist skills required to work in this area [15], underscoring the urgent need for targeted workforce planning and investment in neonatal nursing education and retention.

Consequently, the persistent demand for neonatal care must be met with evidence-based strategies to recruit, develop, and retain a skilled neonatal nursing workforce. Despite the critical role neonatal nurses play in caring for vulnerable newborns, there remains a lack of targeted workforce planning and limited national data on neonatal nursing capacity. To address this gap, the Australian College of Neonatal Nurses Leadership and Research Special Interest Groups convened a national workshop to identify and prioritise strategies for workforce sustainability. This study aimed to explore strategies to strengthen the neonatal nursing workforce in Australia using the World Café methodology. The workshop brought together neonatal nurses, educators, and leaders to collaboratively generate recommendations for stakeholders. The research question guiding this study was: *What strategies can support the recruitment*, *development*, *and retention of neonatal nurses in Australia?*

## 2. Methods

### 2.1. Study Design

To explore strategies for strengthening the neonatal nursing workforce in Australia, we employed the World Café method, a qualitative participatory design that facilitates collaborative dialogue and co-creation of knowledge. This approach was selected over other qualitative methods due to its capacity to generate diverse perspectives in a non-hierarchical, inclusive environment, particularly suited to workforce strategy development.

Originally developed by Brown and Isaacs [16] for organisational change, the World Café has gained traction in qualitative research for its ability to foster mutual learning and produce actionable insights [17]. In this study, the World Café was conducted in a neutral, café-style setting to encourage informal yet focused conversations.

A total of 27 participants including neonatal nurses, educators, and leaders were divided into six groups, each assigned one guiding question from Table 1. These questions were developed through a literature review and refined through discussion among the author team, who are experienced clinicians and academics. While not formally pretested, the questions were reviewed for relevance and clarity prior to the workshop.

The World Café format followed seven key stages: (1) Clarify the Context, (2) Create a Hospitable Space, (3) Explore Questions that Matter, (4) Encourage Everyone’s Contribution, (5) Connect Diverse Perspectives, (6) Listen Together for Patterns and Insights, and (7) Share Collective Discoveries [14,16]. This structured yet flexible approach was particularly well-suited to the aims of the workshop, which sought to generate collaborative, practice-informed strategies for neonatal nursing workforce development. The informal café-style environment fostered open dialogue and psychological safety, encouraging participants to share diverse perspectives grounded in lived experience. The rotating table format, consistent facilitation, and visual documentation on butcher paper enabled the emergence of unanticipated insights and supported inductive reasoning. At the end of each rotation, participants identified key priorities, which were later transcribed into an interactive database, a structured electronic spreadsheet designed to collate and organise participant-generated data for thematic analysis. This method not only facilitated rich data collection but also empowered participants to co-create actionable recommendations, making it an ideal fit for addressing complex workforce challenges in neonatal care.

### 2.2. Participants and Recruitment

Participants were recruited primarily through convenience sampling, targeting individuals who were accessible and met the study’s inclusion criteria. This was complemented by snowball sampling, where initial participants were encouraged to share the study invitation with colleagues and peers. The recruitment strategy aimed to ensure diversity across states and territories, as well as across professional roles and experience levels within neonatal care. Eligible participants included neonatal nurses and midwives who were members of the Australian College of Neonatal Nurses and actively engaged in clinical practice, education, leadership, or research related to neonates, their families, or the neonatal care environment. Additionally, undergraduate nursing and midwifery students with an interest in neonatal nursing were invited to participate.

A total of 27 participants took part in the workshop. Despite extending invitations to 250 undergraduate students via email, no students participated. This may be attributed to the timing of the workshop, which was held on a weekday during the academic semester, potentially conflicting with scheduled classes and clinical placements. The absence of student voices may have limited the generational and educational diversity of perspectives represented in the findings. All participants were female. (See Table 2: Participant demographics). Given the qualitative nature of the study and the structured World Café methodology, the number of participants (*n* = 27) was deemed sufficient to achieve data saturation, as recurring themes and consistent insights emerged across the six discussion groups.

### 2.3. Data Collection

The one-day World Café workshop was held in a neutral, non-clinical meeting space near Sydney Airport, selected to provide a welcoming and accessible environment for participants from across Australia. This setting allowed neonatal nurses, midwives, educators, and leaders to step away from their clinical routines and engage in focused, collaborative dialogue on real-world workforce challenges.

The workshop was structured around six discussion tables, each focused on a guiding question developed through literature review and expert consultation (see Table 1). Each group consisted of four to five participants, and discussions at each table lasted 45 min. After each round, participants rotated between tables to promote cross-pollination of ideas, while facilitators remained at their assigned tables to ensure consistency in guidance and continuity in the discussion process. Facilitators were members of the Research or Leadership Special Interest Groups (SIGs) and were trained to encourage inclusive participation, maintain focus on the guiding question, and support the documentation of insights.

Participants recorded their ideas and recommendations directly onto butcher paper and sticky notes provided at each table, allowing for visual tracking of emerging themes (see Figure 1). At the end of each rotation, groups were asked to identify their top three priorities or ideas, which were then transcribed into an interactive database, a structured electronic spreadsheet designed to collate and organise participant-generated data for thematic analysis [18,19].

The workshop was not audio recorded; instead, the written contributions formed the primary data source. While this approach limited the richness of verbatim dialogue, it ensured that participants’ own words and priorities were captured authentically. These notes were later synthesised to identify key themes and inform recommendations for strengthening the neonatal nursing workforce.

### 2.4. Data Analysis

The World Café method generated rich qualitative data from participants [20]. To analyse this data, we employed thematic analysis following Braun and Clarke’s six-phase framework [21], using an inductive orientation to allow themes to emerge directly from the data rather than being prefigured by the guiding questions. This approach was well-suited to the participatory nature of the study, enabling the identification of patterns grounded in participant experience.

All data generated during the workshop including texts written on butcher paper and sticky notes were transcribed and organised under the initial discussion topics. These transcripts were then manually coded by members of the research team over several weeks through a series of virtual meetings, followed by a final face-to-face session to reach consensus. Coding was conducted without the use of qualitative software to allow for close engagement with the data.

To enhance rigour, all researchers involved in the analysis held doctoral qualifications, were experienced in qualitative research, and had extensive neonatal clinical expertise. Importantly, while some members of the research team attended the workshop as facilitators, they did not participate in the discussions, thereby maintaining analytical independence. Intercoder agreement was achieved through collaborative discussion, where initial codes were reviewed and refined into broader themes and subthemes.

The final workshop findings were categorised into four key themes: (1) Leadership and Advocacy, (2) Professional Identity, (3) Future Vision and Innovation, and (4) Creating a Supportive Culture. These themes emerged organically from the data and provided a framework for nursing leaders and stakeholders to address workforce challenges in neonatal care.

### 2.5. Ethical Approval and Informed Consent

This research project was approved by Western Sydney University Human Ethics number: H15910. Participant information sheets with the researcher’s details and contact information including email and phone numbers were provided. Participation was completely voluntary. Participants were reassured that information would be kept anonymous and confidential and advised they could withdraw from the study at any time to the point of publication. No financial incentives were offered for involvement in the study. All participants provided written consent.

To preserve anonymity, privacy, and confidentiality, identifying information was stored separately from the main dataset. The World Café methodology, as a participatory qualitative approach, was guided by both the research team and participants, with results formulated collaboratively during the workshop. Aggregated findings were presented without identifying individual contributors. Discussions were not audio recorded; instead, data were captured through participant-generated notes on butcher paper and sticky notes, which were later transcribed and analysed.

Member checking occurred both during the workshop, through real-time review and discussion of emerging ideas and post-analysis, when the research team revisited the themes to ensure they accurately reflected participant contributions. This process enhanced the credibility and trustworthiness of the findings.

The research team consisted of experienced neonatal nurses and midwives, including three academics and one clinician-researcher. While some team members had pre-existing professional relationships with participants and acted as facilitators during the workshop, they did not contribute to the discussions, maintaining analytical independence. The team practiced reflexivity, acknowledging their positionality and potential biases, and actively encouraged participants to lead the dialogue and generate ideas.

Rigour was maintained through transparent and systematic processes aligned with qualitative research standards. The World Café workshop was designed to promote credibility (through member checking and inclusive facilitation), dependability (via consistent procedures and documentation), confirmability (through collaborative coding and consensus-building), and transferability (by clearly describing the context and methodology). However, the inclusion of only Australian participants, many of whom held leadership or education roles, may limit the transferability of findings to other healthcare contexts or more diverse workforce segments.

## 3. Results

### 3.1. Theme 1: Advocacy and Leadership

World Café attendees emphasised the critical role of advocacy and leadership in the development and sustainability of the neonatal nursing workforce. Participants described key aspects of these goals under two main themes being resources and safety. Participants identified a neonatal nurse’s ability to access a safe working environment where they felt valued and capable of advancing their careers was incumbent upon nursing leadership and adequate resource allocation.

Investment in resources, such as dedicated nurse educators and ongoing training programmes, are essential for maintaining high standards of care and supporting continuous learning. Participants discussed the impact of “*resource heavy transition programs*” and “*the complexity of addressing staffing shortages that ensures adequate coverage*.” Participants highlighted the need to pull together resources to stabilise the current workforce, by stating, “*The struggle isn’t employing new nurses*, *it is keeping them*.” Participants suggested that work-based career advancement opportunities might be facilitated by “*encouraging staff movement*” and “*developing a talent pool*.” Factors such as “*improving staff facilities*” and “*providing non-clinical time for professional development and personal well-being*,” were essential for maintaining a motivated and skilled workforce.

Attendees underscored the need for leaders who can inspire and mentor staff. Leadership styles that encourage an “open door” policy, regular check-ins, and constructive feedback within a psychological safety framework are foundational to a resilient, supportive work culture. Leaders need to advocate for these resources otherwise this leads to sub-optimal staffing, burnout, and impaired healthcare quality. Nurses often wonder, “*Am I able to provide great care?*” and “*Am I safe?*” Advocacy and leadership are crucial in addressing these concerns by ensuring the availability of human and material resources to bridge education gaps, improve nursing skills, and maintain care standards.

Participants identified the importance of governing bodies such as Neonatal Organisations in advocating and leading change “*Using our professional voices*” to enhance the profile and celebrate the neonatal nursing profession. Participants thought this could be achieved through developing resources that assist with the implementation of neonatal nursing standards, role modelling best practice, inspiring mentorship, and leadership opportunities through innovative professional development opportunities. Increasing the visibility of neonatal professional organisations representing neonatal nurses was also discussed as many felt as individuals their voice was silent but as a group (college) they can be influential.

### 3.2. Theme 2: Professional Identity and Engagement

The theme of professional identity and engagement emerged directly from the data, with participants consistently highlighting its importance in boosting the profile and morale of neonatal nurses. Through thematic analysis of the World Café workshop discussions, two subthemes were identified: professional profile and public perception.

Participants frequently referred to the need to “*raise the profile of neonatal nursing*” and “*how we profile ourselves*,” underscoring the significance of a strong professional identity. This identity helps neonatal nurses better understand their roles within their specialised practice area and broader organisational responsibilities. It is shaped by individual perceptions, clinical experiences, and educational pathways, and continues to evolve throughout a nurse’s career.

A clearly articulated and adaptable **professional identity** was seen as essential for delivering high-quality care, attracting future students, and retaining skilled staff. Participants noted that neonatal nursing practice is dynamic, requiring clinical expertise, critical thinking, and flexibility. As such, a strong professional identity supports nurses in navigating changing practice demands and maintaining confidence in their scope of practice.

The second subtheme, ***public perception***, was also strongly emphasised. Participants expressed concern that commonly used terms like “*nursery*” fail to reflect the complexity and acuity of neonatal nursing. They advocated for the use of precise terminology to convey the specialised nature of care provided to vulnerable neonates. Using “*our professional voices*” to “*highlight neonatal outcomes*” was identified as a key strategy to enhance public understanding and appreciation of the profession.

In summary, a strong professional identity shaped by internal clarity and external advocacy was viewed as a foundational element in building workforce resilience among neonatal nurses.

### 3.3. Theme 3: Future Vision and Innovation

Imagining a future-ready neonatal care environment begins with a workforce that is not only highly skilled and knowledgeable but also professionally fulfilled and motivated. This vision, shaped by rich workshop discussions, revealed three interrelated priorities: enhancing education and learning modalities, expanding clinical exposure and inter-professional collaboration, and embracing technology and innovation in training. These ideas emerged organically from participant contributions and were collectively prioritised as essential strategies for advancing neonatal nursing.

Participants strongly advocated for investment in neonatal nurse education as a cornerstone for improving care quality and ensuring workforce sustainability. A central question, “*How do we change quantity into quality in education practices?*”, sparked a range of suggestions aimed at enhancing learning experiences. These included simulation-based training, gamified learning environments, and AI-driven platforms that support self-directed and experiential learning. Blending traditional hands-on approaches such as “*see one*, *do one*, *teach one*” with digital tools was seen as vital for accommodating diverse learning styles. This hybrid model reflects a generational shift, particularly among millennial nurses, who value collaborative, tech-enabled learning environments that foster peer engagement and shared growth.

Participants also emphasised the importance of meaningful clinical exposure to prepare neonatal nurses for real-world practice. Strategies included embedding “*teachable moments*” and structured secondments into neonatal units and retrieval services during undergraduate and postgraduate training. These experiences were viewed as critical for deepening clinical insight and fostering inter-professional collaboration. Standardised postgraduate programmes co-developed with neonatal care units were recommended to ensure consistency and relevance in training. Experiential learning tools such as mock codes, role play with trained actors, and case-based discussions were highlighted for their ability to strengthen team dynamics and translate theoretical knowledge into practice. Incorporating consumer feedback was also seen as a valuable strategy to enrich learning and promote reflective practice.

Technology was identified as a powerful enabler of accessible, engaging, and inclusive education. Participants proposed using social media, video-guided discussions, interactive quizzes (e.g., Kahoot), and QR code-linked tutorials to enhance learning particularly for nurses in rural, remote, and resource-limited settings. Creative approaches such as escape rooms were suggested as novel ways to debrief, explore human factors, and communicate practice changes in informal, team-building contexts. These ideas reflect a growing demand for flexible, digitally enhanced education that supports collaboration and continuous learning. Comments such as “More simulation and video resources around complex situations…Collaboration!!” captured the collective enthusiasm for technology-driven solutions that strengthen multidisciplinary teams and improve care outcomes.

In summary, participants envisioned a neonatal care environment defined by innovation, collaboration, and educational excellence. Strengthening professional development through diverse learning modalities, meaningful clinical exposure, and technology-enhanced training was seen as essential for cultivating a resilient, future-ready neonatal nursing workforce. This vision reflects both professional aspirations and generational shifts toward digital collaboration, positioning neonatal nurses as leaders in shaping the future of healthcare.

### 3.4. Theme 4: Creating a Supportive Culture

A significant theme that emerged throughout the workshop was the need to cultivate a supportive culture within neonatal nursing environments. Participants posed reflective questions such as, “*How can we support the next generation of nurses?*” and “*What resources do we give them?*” highlighting a collective desire to counter resistance and enhance motivation for change. Worksheets captured concerns around entrenched workplace behaviours, with prompts like “*How can I help the staff on the floor [clinical nurses] change culture?*” and “*How do you get staff who are rooted in place to accept change and help new staff?*”

Participants expressed feeling “*hopeful*, *excited and motivated*” about the potential for progress, suggesting it is “*time to get things done.*” They emphasised the importance of “*supportive networks*, *celebrating and encouraging efforts*,” especially for new employees and those transitioning into advanced roles requiring higher-order skills and attributes. Sharing positive feedback was seen as a catalyst for “*growth and motivation*,” contributing to increased job satisfaction, “*self-worth and happiness.*” These ideas were grouped under a broader theme of Motivation and Wellbeing, where recognition, encouragement, and a sense of belonging were viewed as essential to sustaining morale.

Participants also stressed the importance of Resilience and Support through effective mental health and wellbeing initiatives. Neonatal nurses want to “feel safe,” “*protected and nurtured*,” and supported through emotionally demanding situations. The need to “*reconnect with empathy*” was highlighted, alongside calls for “*formal*, *timely*, *regular*, *safe and accessible*” debriefing sessions. Training specific staff to lead these sessions and respond to acute incidents was considered vital. Prompt debriefing after critical events was seen as a way to prevent emotional overload and foster psychological safety.

Team-building activities and authentic support from colleagues and management were repeatedly mentioned as strategies to build a resilient and cohesive workforce. Participants proposed planned “*social activities*,” such as monthly Saturday outings, and celebrating team and individual successes to enhance camaraderie. Improving staff facilities and providing non-clinical time for professional development and personal wellbeing were also considered important for promoting “*work–life balance.*”

Effective rostering and manageable workloads were identified as core burnout prevention strategies. Participants openly discussed challenges such as “*work anxiety*”, “*teaching/mentoring/preceptor fatigue*,” and “*unrealistic expectations*,” which often led to feelings of guilt and frustration. These pressures sometimes resulted in task-based care and self-doubt, with nurses asking, “*How can I do it all?*”

An insightful subtheme emerged around Generational Dynamics, where participants noted differences in work ethics and expectations across age groups. Managing a multi-generational workforce while optimising skill mix and providing tailored support was seen as essential for staff retention. One participant reflected, “*Younger nurses want to collaborate and learn from each other*, *sharing is how they grow.*” This generational shift calls for leadership strategies that embrace digital learning, flexible communication styles, and inclusive team-building approaches. Modelling positive behaviours and “*calling out and counteracting negativity*” were suggested as ways to foster a respectful and growth-oriented culture.

Workplaces that cultivate curiosity, encourage open communication, celebrate successes, and plan social events were viewed as genuinely supportive and capable of building strong, resilient nursing teams.

In summary, participants envisioned a neonatal nursing culture grounded in empathy, recognition, and psychological safety. Motivation, wellbeing, and resilience were seen as interdependent pillars of a supportive workplace. Addressing generational differences and fostering inclusive leadership were considered key to sustaining a thriving workforce.

We adopted an innovative and inclusive approach to exploring neonatal nursing workforce challenges through the World Café methodology. This facilitated rich, collaborative dialogue among participants, generating grounded insights and practical strategies. The process enabled us to surface four interconnected themes: **Professional Identity and Engagement, Education, Exposure, and Innovation, Leadership and Future Visioning, and Supportive Culture and Workforce Resilience**, each informed by authentic participant narratives and lived experiences. These themes collectively offer a strategic framework for strengthening the neonatal nursing workforce. The recommendations derived from these discussions are intended to guide future workforce development and are shared with the neonatal nursing community and broader stakeholders for reflection and action (see Table 3: exemplar of how theme and subtheme inform recommendations).

## 4. Discussion

Leadership and advocacy emerged as pivotal themes throughout the workshop, with participants emphasising their importance in building a strong and sustainable neonatal workforce. Comments reflected a collective desire for leaders who champion staff development and foster positive change. This aligns with Kiwanuka et al. [22], who found that active leaders sharing a common vision and advocating for their teams are highly valued. Authentic and transformational leadership styles were noted to create vibrant work environments and support staff retention [22].

The urgency of leadership development is underscored by the projected retirement of a quarter of Australia’s neonatal workforce within the next decade [23,24]. Participants stressed the need for quality neonatal education that integrates staff wellbeing and leadership capacity. This is particularly relevant given the mandatory registration standards under the Health Practitioner Regulation National Law 2009, which require Australian registered nurses and midwives to maintain recency of practice and engage in continuing professional development [25]. Professional advancement, including the awarding of advanced practice roles and postgraduate qualifications, is closely tied to these standards [26].

Participants also highlighted the need to modernise neonatal education to meet evolving clinical demands. Emerging technologies such as simulation, artificial intelligence (AI), gamification, and social media were seen as transformative tools that enhance training and mitigate latent safety threats. A systematic review of 50 studies involving 7536 registered nurses across 19 countries identified four key domains for neonatal nursing knowledge: standard care, intensive care, end-of-life care, and family-centred care [25]. Techniques ranging from low- to high-fidelity simulation and virtual reality (VR) were praised for offering immersive, realistic learning experiences that address communication breakdowns and human factors [27,28,29]. Participants in our study echoed these findings, advocating for blended educational approaches that combine practical tasks, simulation, and role-play to promote multidisciplinary collaboration and improve care quality [27,28,29].

The theme of professional identity was also strongly emphasised. Participants consistently expressed that a robust professional identity is essential for neonatal nurses to understand their roles within both their specialised practice and the broader organisational context. One participant noted, “*We need to be clear about who we are and what we bring to the team.*” This clarity was seen as foundational for effective teamwork, high-quality care, and personal fulfilment.

Roerink [28] found that professional identity significantly influences nurses’ ability to deliver quality care and impacts emotional labour and perceived organisational justice [29]. A well-defined professional identity fosters pride, motivation, and resilience, qualities that are critical for job satisfaction and retention. As one participant shared, “*When you know your value*, *you show up differently.*” Watson [29,30,31,32] similarly concluded that nurturing nurses’ professional identity enhances their sense of purpose and may reduce attrition.

Creating supportive working environments is important as psychological and physical stress and burnout are common among neonatal nurses which may result in mental fatigue, exhaustion, low morale, high staff absenteeism and staff leaving the profession [33]. Nurses exposed to repeated and ongoing stress are at greater risk of experiencing burnout [33].

The World Health Organization [34] recognises burnout as a serious health issue affecting an individual’s mental wellbeing and may result from inadequately managed chronic workplace stress. Evidence suggests an estimated 25–50% neonatal staff are affected by burnout symptoms [33]. Contributing factors include perceived stress, socioeconomic factors, and a lack of social support [33], along with workloads, competing demands when caring for critically ill infants, emotional competence when working with families and continuous changes in technology. Moral distress, anxious attachment and shame-proneness have been identified as moderate to high correlations to burnout in neonatal nurses [24,35]. Participants in our study questioned whether they were able to provide safe care and report leadership styles that support staff is crucial. Similarly, research conducted by Barr [36] suggest managers who actively seek to minimise staff moral distress from compromised care, futile care and untruthful care have also been reported to assist in reducing staff burnout. Additionally, Staples et al. [24] suggest staff wellbeing needs to be incorporated in the education of neonatal nurses, as failure to consider staff wellbeing may negatively impact on delivering safe quality healthcare to neonates and their families [24].

A study exploring resilience amongst neonatal nurses indicates interventions such as achieving personal accomplishments in addition to managerial support, and collegial relationships results in more resilient nurses and reduced perceived burnout [37]. Similarly, a scoping review conducted by Sihvola et al. [38] suggests communication, safe and supportive environments and relational leadership styles were found to support nurse resilience. Nurses are reported to experience adversity at interpersonal, cultural and systems levels while contextual challenges such as heavy workload, communication and working in partnership with families along with individual factors, and social networks are considered critical to building resilience [39]. Valdez [40] suggests building resilience alone is not enough, and promoting nurse wellness and preventing harm requires organisational and multi system-level change to ensure nurses practice in safe and healthy environments.

Grunberg et al. [41] identified a lack of integrated psychosocial support programmes for neonatal staff and noted that researchers and clinicians often operate in silos, underscoring the need for collaborative and innovative strategies to enhance staff wellbeing. Similarly, data from our World Café workshop revealed a strong consensus among participants regarding the urgent need for programmes that prioritise both the mental and physical wellbeing of neonatal nurses. Comments such as “*We need to feel safe and supported to do our best work*” and “*It’s time to invest in the people behind the care*” reflected a shared desire for peer support networks, recognition initiatives, and wellbeing-focused interventions.

Despite this clear demand, participants also acknowledged several barriers to implementing such programmes. These included limited funding, competing service priorities, and inconsistent leadership support. The informal nature of psychosocial care, often viewed as secondary to clinical duties was seen as a challenge to formalising and resourcing these initiatives. Participants emphasised that without leadership commitment and structural investment, efforts to support staff wellbeing risk being unsustainable or tokenistic.

These findings reinforce the need for systemic change, where wellbeing is embedded into workforce planning and leadership development. Establishing peer-led support networks, celebrating contributions through recognition programmes, and integrating wellbeing into professional development pathways were proposed as actionable strategies to foster a more resilient and valued neonatal nursing workforce.

Our study highlighted the importance of supporting a multi-generational neonatal workforce, echoing findings by Hisel [42], who emphasised the need to engage nurses across generational cohorts. Generational differences shaped by historical events, technological advances, cultural shifts, and varying communication styles were seen to influence workplace engagement and motivation. Hisel [42] found that Veteran (1925–1945) and Baby Boomer (1946–1965) nurses reported higher levels of engagement compared to Generation X (1966–1980), Millennials (1981–1995), and Generation Z (1996–2010), with the latter groups scoring lowest in workplace engagement.

Participants in our study expressed concern about the implications of these differences for workforce sustainability. With many experienced senior staff nearing retirement, mentoring and knowledge transfer to younger nurses were viewed as essential strategies to preserve institutional knowledge and foster leadership continuity. Comments such as “*We need to pass on what we know before it’s lost*” and “*Younger nurses want to collaborate and learn from each other—sharing is how they grow*” reflected a desire to bridge generational gaps through inclusive leadership and tailored support.

Prioritising engagement across age groups is critical for retention. Leadership strategies that acknowledge generational preferences such as flexible learning formats, recognition of diverse work motivations, and opportunities for intergenerational collaboration can help build a cohesive and resilient neonatal workforce.

Additionally, our findings underscore the critical need for neonatal nurses and leaders to take proactive steps in cultivating a resilient workforce, one that supports recruitment, retention, and the delivery of safe, high-quality care for neonates and their families. This need was further validated through a broader consultation with Australian neonatal nurses at a national neonatal nursing conference, where 110 attendees contributed 285 responses to the question, “*What do you need to succeed in your neonatal nursing career?*” The most frequently cited needs included support, education, mentorship, leadership, flexibility, and recognition. These responses strongly align with the themes identified in our study and reinforce the urgency of implementing workforce strategies that are responsive to the lived experiences and aspirations of neonatal nurses.

### 4.1. Recommendations

Achieving sustainable improvements in neonatal nursing workforce development requires a coordinated, multi-level response. Drawing on the findings from our World Café workshop and subsequent validation at a national neonatal nursing conference, we propose a series of evidence-informed recommendations (see Table 4), organised by stakeholder level to enhance clarity and implementation.

At the **systemic level**, professional nursing organisations and policy makers are encouraged to lead national advocacy campaigns aimed at improving resource allocation, working conditions, and public awareness of neonatal nursing as a specialised field. This recommendation is grounded in participants’ calls to “raise the profile of neonatal nursing” and “use our professional voices.” Leadership development initiatives, including mentorship programmes and structured workshops, should be funded to identify and nurture future leaders, particularly considering the projected retirement of senior neonatal staff. To support innovation and education, nursing associations should offer grants for technology-integrated learning—such as artificial intelligence, simulation, and gamification—reflecting participants’ interest in modernising educational modalities. Additionally, programmes focused on mental and physical wellbeing, peer support networks, and recognition of contributions should be established to address the expressed need for “supportive networks” and a sense of being valued.

At the **managerial level**, nurse leaders and unit managers play a critical role in shaping workplace culture and supporting staff wellbeing. Regular leadership rounds, open feedback channels, and inclusive communication practices are recommended to build trust and engagement across generational cohorts. Mentorship programmes that pair experienced nurses with newer staff, alongside clearly defined roles and personalised professional development plans, can strengthen professional identity and retention. Flexible work practices, including adaptable scheduling and protected non-clinical time, are essential to address burnout and promote work–life balance. Managers should also invest in simulation-based learning and online education platforms to support continuous learning and foster team-based care.

At the **individual level**, clinical nurses are encouraged to engage in advocacy training, form peer-led groups to address staffing and practice issues and take on leadership roles within their units. These actions reflect participants’ desire to “help the staff on the floor change culture.” Nurses should also pursue professional growth through public speaking, reflective practice groups, and contributions to research and education, thereby fostering pride, purpose, and career progression. Participation in peer support initiatives, celebration of team successes, and engagement in personal wellbeing activities are further recommended to cultivate a positive and resilient workplace culture.

These recommendations are grounded in the lived experiences and insights of neonatal nurses and leaders. They offer a strategic framework for strengthening workforce sustainability, professional identity, and care quality across the neonatal sector.

### 4.2. Limitations

This study was conducted with a specific cohort of neonatal nurses who were members of the Australian College of Neonatal Nurses (ACNN). The participant demographics predominantly comprised women aged between 40 and 60 years, with no representation from newly graduated neonatal nurses, nursing students, or male clinicians. Due to the nature of the World Café methodology, which involved anonymous contributions via sticky notes on butcher paper, individual participant identifiers were not assigned. While this approach facilitated open and inclusive dialogue, it limits the ability to attribute specific insights to particular demographic groups.

Importantly, the findings may not be generalised beyond the neonatal nursing population. The targeted focus on neonatal nurses provided rich, context-specific insights; however, transferability to other nursing specialties or broader healthcare settings may be limited. Future research should aim to include a more diverse sample, encompassing a wider range of career stages, genders, and professional backgrounds to enhance representativeness and applicability.

## 5. Conclusions

This study highlights the shared responsibility of nursing associations, nurse managers, and clinical nurses in strengthening advocacy, leadership, professional identity, innovation, and support within the neonatal nursing workforce. Through the application of the World Café methodology, we provided a step-by-step account of how collaborative dialogue was used to surface workforce challenges and co-create actionable strategies. The findings offer a nuanced understanding of the lived experiences of neonatal nurses and present a comprehensive framework for workforce development.

Importantly, the recommendations generated from this study have direct implications for policy and practice. They provide a foundation for nursing organisations and healthcare leaders to inform strategic planning, guide investment in leadership and education initiatives, and shape policies that promote workforce sustainability and retention. By aligning workforce strategies with the voices of neonatal nurses, this study contributes to the development of responsive, evidence-informed policies that support both staff wellbeing and the delivery of high-quality care to neonates and their families.

## Figures and Tables

**Figure 1 healthcare-13-02976-f001:**
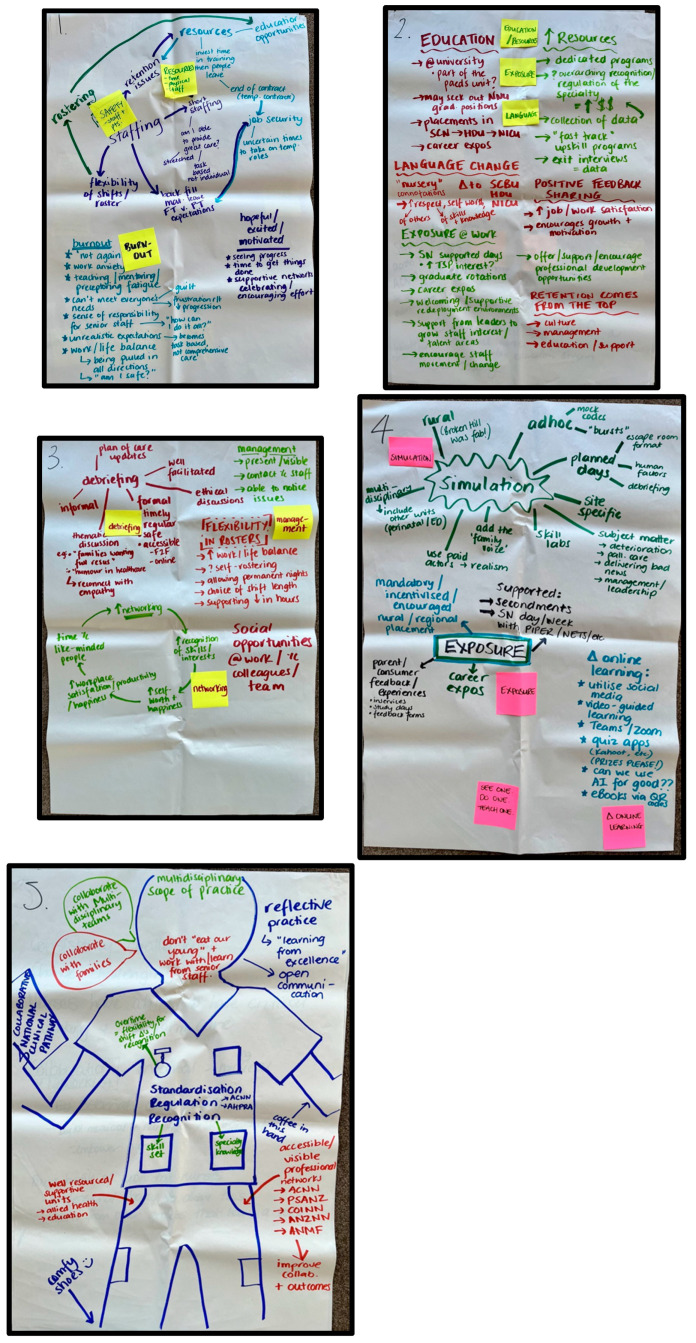
Participant Data.

**Table 1 healthcare-13-02976-t001:** World Café Questions.

1. Given the current changes and challenges facing us in the neonatal workforce, what questions are you asking yourself and how does it impact on your current role? (education, manager, leadership, clinician, researcher)
2. What ideas do you have for getting nurses interested in neonatal nursing and, once we have them, keeping them super happy and engaged?
3. How can we cultivate a culture of resilience and support to enhance the mental health and well-being of neonatal nurses, recognising the unique emotional challenges they face in their daily work? Neonatal nursing has its share of tough moments. What are some battle-tested strategies you’ve come up with or experienced to overcome challenges, big or small?
4. How might we re-envision neonatal nursing education to be more engaging and adventurous, moving away from traditional lectures? What innovative approaches or unconventional ideas could we explore to make learning in this field more exciting and impactful? Any wild ideas?
5. How can we transform neonatal nursing into a dynamic, ever-evolving field by instilling a culture of curiosity, collaboration, and continuous learning among our practitioners? Imagine neonatal nurses as superheroes collaborating with other healthcare heroes. What kind of superhero team-ups would make our neonatal care unbeatable?
6. Picture the dream neonatal nursing environment–what would it look like, sound like, and feel like? How can we make that dream a reality?

**Table 2 healthcare-13-02976-t002:** Participant demographics.

Age	21–30 0	31–40 7	41–50 10	51–60 8	>60 2	-	TOTAL 27
Year’s neonatal experience	0–5 2	6–10 3	11–15 2	16–20 6	21–25 3	>25 11	TOTAL 27
Substantive role (Full Time Equivalent [FTE])	Clinical 10.5 FTE	Education 4.5 FTE	Management 7 FTE	Research 4 FTE	Other 1 FTE	-	TOTAL 27 FTE
Years to retirement	0 1 (recently retired)	1–2 2	3–5 3	6–10 5	11–15 2	>15 14	TOTAL 27
State/ Territory	QLD 6	NSW 13	ACT 3	VIC 5	TAS, SA, WA 0	NT 0	TOTAL 27

**Table 3 healthcare-13-02976-t003:** Exemplar of how theme and subtheme inform recommendations.

Theme	Subthemes	Recommendations
Leadership and Advocacy	Resources Safety	We recommend a sustained and increased investment in the material and human resources needed to make nurses feel safe and able to advance their careers. Doing so enables nurses to bridge education gaps, improve nursing skills, and improve care standards.
Professional Identity and Engagement	Professional profile Public perception	Nurses must be able to access the education and clinical experience required to develop their professional identities, as these are crucial for advancing a positive public perception of neonatal nursing and enabling the high-quality care provision known to encourage staff recruitment and secure staff retention.
Future Vision and Innovation	Quality Exposure Collaboration	The avenues to increase clinical exposure must be explored, and the various education methods used to build individualised nursing knowledge, skill, and inter-professional team harmony, pursued.
Creating a Supportive Culture	Motivation Wellbeing Flexibility	Leadership styles that encourage flexibility, positive motivation, and team psychological safety must be employed to build a resilient workforce, meet the needs of multi-generational staff members, and improve patient healthcare quality and safety.

**Table 4 healthcare-13-02976-t004:** Recommendations.

**Lead the delivery of high-quality neonatal care**
Prioritise sustained and increased investment in both material and human resources. Advocate to ensure nurses feel safe, supported, and empowered to advance their careers, we can bridge educational gaps, enhance nursing skills, and elevate care standards.
**Use our professional voices**
Highlight “neonatal outcomes” as a powerful strategy to elevate public understanding of the vital work performed by neonatal nurses, positively influence public perceptions of the profession, and create an environment that attracts and retains skilled staff.
**Engage in novel thinking that considers nurses working at their full scope of practice**
Exploring innovative avenues to increase clinical exposure and adopting diverse educational methods tailored to individual learning needs. Focus on building personalised nursing knowledge, skills, and fostering inter-professional team harmony, we can prepare the workforce for future challenges and ensure the continuous evolution of neonatal care practices.
**Develop a supportive and resilient workforce**
Flexibility, positive motivation, and psychological safety are crucial for creating a supportive and resilient work environment. Meet the diverse needs of a multi-generational workforce, we can enhance staff wellbeing, improve retention, and ultimately elevate the quality and safety of patient care within neonatal nursing.

## Data Availability

Data may be available upon request through the corresponding author and upon authorisation of the Western Sydney University Australia Human Research Ethics Committee.

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
