# Peer review of "Strengthening the Neonatal Workforce Through World Café Methodology"

_healthcare, 2025, doi:10.3390/healthcare13222976_

Round 1

Reviewer 1 Report

Comments and Suggestions for Authors

This is a well-structured qualitative study employing the World Café methodology to explore strategies for recruitment and retention in neonatal nursing. The paper addresses a highly relevant issue—neonatal workforce sustainability—and provides valuable insights for nursing leaders, educators, and policymakers. The manuscript aligns well with Healthcare’s focus on applied health sciences and professional development.

Author Response

Please see attached- I am unable to submit thte SRQR document here. It will attached as Supp file

Reviewer 2 Report

Comments and Suggestions for Authors

Dear Authors: The manuscript makes a valuable contribution to neonatal workforce development literature and is suitable for publication following moderate revision to strengthen data integration, reduce repetition, and enhance analytical interpretation.

Analytic report is provided to authors for clear suggested revision.

Respectfully

The Reviewer
